# Black-Box Dissector: Towards Erasing-based Hard-Label Model Stealing Attack

## Abstract

Previous studies have verified that the functionality of black-box models can be stolen with full probability outputs. However, under the more practical hard-label setting, we observe that existing methods suffer from catastrophic performance degradation. We argue this is due to the lack of rich information in the probability prediction and the overfitting caused by hard labels. To this end, we propose a novel hard-label model stealing method termed *black-box dissector*, which consists of two erasing-based modules. One is a CAM-driven erasing strategy that is designed to increase the information capacity hidden in hard labels from the victim model. The other is a random-erasing-based self-knowledge distillation module that utilizes soft labels from the substitute model to mitigate overfitting. Extensive experiments on four widely-used datasets consistently demonstrate that our method outperforms state-of-the-art methods, with an improvement of at most $8.27\%$. We also validate the effectiveness and practical potential of our method on real-world APIs and defense methods. Furthermore, our method promotes other downstream tasks, *i.e.*, transfer adversarial attacks.

## 1 Introduction

Machine learning models deployed on the cloud can serve users through the application program interfaces (APIs) to improve productivity. Since developing these cloud models is a product of intensive labor and monetary effort, these models are valuable intellectual property and AI companies try to keep them private. However, the exposure of the model's predictions represents a significant risk as an adversary can leverage this information to steal the model's functionality, *a.k.a.* model stealing attack [22, 20, 21]. With such an attack, adversaries are able to not only use the stolen model to make a profit, but also mount further adversarial attacks [34, 29]. Besides, the model stealing attacks is a kind of black-box knowledge distillation which is a hot research topic. Studying various mechanisms of model stealing attack is of great interest both to AI companies and researchers.

Previous methods [20, 34, 21] mainly assume the complete probability predictions of the victim model available, while the real-world APIs usually only return partial probability values (top-$k$ predictions) or even the top-1 prediction (*i.e.*, hard label). In this paper, we focus on the more challenging and realistic scenario, *i.e.*, the victim model only outputs the hard labels. However, under this setting, existing methods suffer from a significant performance degradation, even by 30.50% (as shown in the Fig. 1 (a) and the appendix Tab. I).

To investigate the reason for the degradation, we evaluate the performance of attack methods with different numbers of prediction probability categories available and hard labels as in Fig. 1 (b). With the observation that the performance degrades when the top-$k$ information missing, we conclude that the top-$k$ predictions are informative as it indicates the similarity of different categories or multiple objects in the picture, and previous attack methods suffer from such information obscured

by the top-1 prediction under the hard-label setting. It motivates us to re-mine this information by eliminating the top-1 prediction. Particularly, we design *a novel CAM-based erasing method*, which erases the important area on the pictures based on the substitute model's top-1 class activation maps (CAM) [24, 33] and queries the victim model for a new prediction. Note that we can dig out other class information in this sample if the new prediction changes. Otherwise, it proves that the substitute model pays attention to the wrong area. Then we can align the attention of the substitute and the victim model by learning clean samples and the corresponding erased samples simultaneously.

Besides, previous works on the self-Knowledge Distillation (self-KD) [15], calibration [8], and noisy label [31] have pointed out the hard and noisy labels will introduce overfitting and miscalibration. More specifically, the attack algorithms cannot access the training data, and thus can only use the synthetic data or other datasets as a substitute, which is noisy. Therefore, the hard-label setting will suffer from overfitting, which leads to worse performance, and we verify it by plotting the loss curves in Fig. 1 (c). To mitigate this problem, we introduce *a simple self-knowledge distillation module with random erasing (RE)* to utilize soft labels for generalization. Particularly, we randomly erase one sample a certain number of times, query the substitute model for soft-label outputs, and take the average value of these outputs as the pseudo-label. After that, we use both hard labels

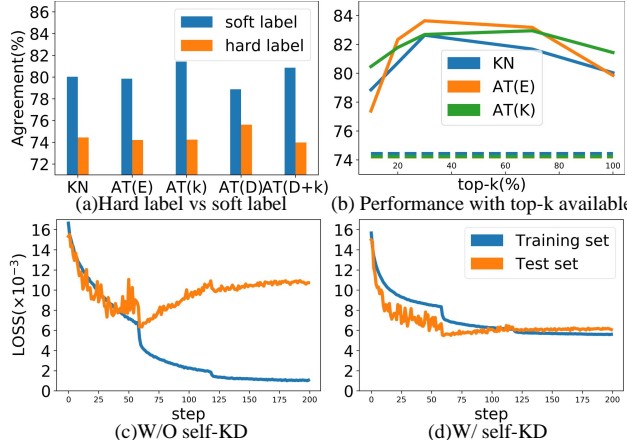

Figure 1: (a) The test accuracies of previous methods with hard labels are much lower than the ones with soft labels. (KN: KnockoffNets, 'AT': ActiveThief, 'E': entropy, 'K': k-Center, 'D': DFAL) (b) The performance decreases as the number of available classes decreases (dotted line : hard-label setting). (c) & (d) Loss curves for training/test set during model training without and with self-KD. All results are on the CIFAR10 dataset.

from the victim model and pseudo labels from the previous substitute model to train a new substitute model. Therefore, we can also consider the ensemble of the two models as the teacher in knowledge distillation. As in Fig. 1 (d), such a module helps generalization and better performance.

In summary, we propose a novel model stealing framework termed *black-box dissector*, which includes a CAM-driven erasing strategy and a RE-based self-KD module. Our method is orthogonal to previous approaches [20, 21] and can be integrated with them. The experiments on four widely-used datasets demonstrate our method achieves $43.04 - 90.57\%$ test accuracy ($47.60 - 91.37\%$ agreement) to the victim model, which is at most $8.27\%$ higher than the state of the art method. We also proved that our method can defeat popular defense methods and is effective for real-world APIs like services provided by Amazon Web Services (AWS). Furthermore, our method promotes downstream tasks, *i.e.*, transfer adversarial attack, with $4.91\% - 16.20\%$ improvement.

## 2 Background and Notions

**Model stealing attack** is aim to find a substitute model $\hat{f}\colon [0,1]^d \mapsto \mathbb{R}^N$ that performs as similarly as possible to the black-box victim model $f\colon [0,1]^d \mapsto \mathbb{R}^N$ (with only outputs accessed). Papernot et al. [22] first observed that online models could be stolen through multiple queries. After that, due to the practical threat to real-world APIs, several studies paid attention to this problem and proposed many attack algorithms.

These algorithms consist of two stages: 1) constructing a transfer dataset $D_T$ (step 1 in Fig. 2) and 2) training a substitute model. The transfer dataset is constructed based on data synthesis or data selection and then feed into the victim model for labels. Methods based on data synthesis [34, 14, 2] adopt the GAN-based models to generate a virtual dataset. And the substitute model and the GAN model are trained alternatively on this virtual dataset by querying the victim model iteratively. The data selection methods prepare an attack dataset as the data pool, and then sample the most informative

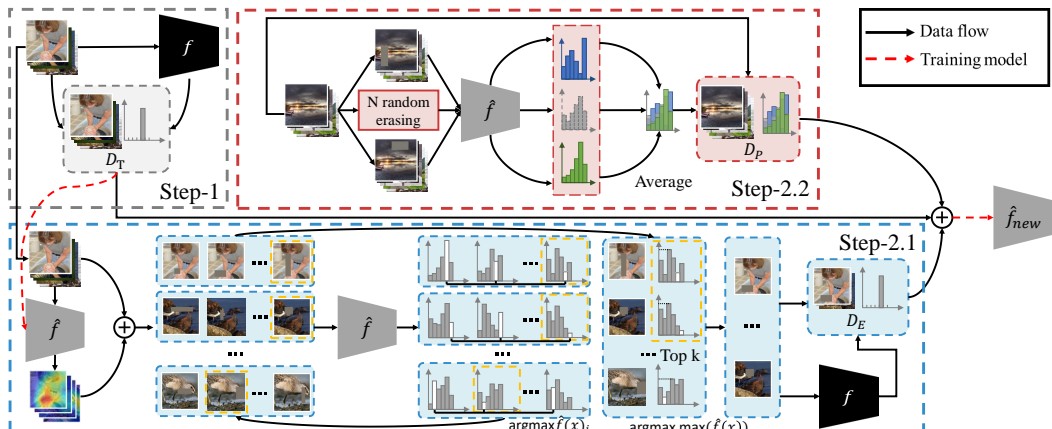

Figure 2: Details of our proposed black-box dissector with a CAM-driven erasing strategy (step 2.1) and a RE-based self-KD module (step 2.2). In step 2.1, the images in transfer set $D_T$ are erased according to the Grad-CAM, and we selected the erased images with the largest difference from the original images according to the substitute model's outputs. In step 2.2, we randomly erase the unlabeled image $N$ times, and then average the outputs of the $N$ erased images by the substitute model as the pseudo-label.

data via machine learning algorithms, *e.g.*, reinforcement learning [20] or active learning strategy [21], uncertainty-based strategy [17], k-Center strategy [25], and DFAL strategy [5]. Considering that querying the victim model will be costly, the attacker usually sets a budget on the number of the queries, so the size of the transfer dataset should be limited as well. Previous methods assume the victim model returns a complete probability prediction $f(x)$, which is less practical.

In this paper, we focus on a more practical scenario that is about hard-label $\phi(f(x))$ setting, where $\phi$ is the truncation function used to truncate the information contained in the victim's output and return the corresponding one-hot vector:

$$\phi(f(x))_i := \begin{cases} 1 & \text{if } i = \arg\max_n f(x)_n\,; \\ 0 & \text{otherwise}\,. \end{cases} \tag{1}$$

With the transfer dataset, the substitute model is optimized by minimizing a loss function $\mathcal{L}$ (*e.g.*, cross-entropy loss function):

$$\begin{cases} \mathbb{E}_{x \sim \mathcal{D}_T}\left[\mathcal{L}\big(f(x), \hat{f}(x)\big)\right], & \text{for soft labels;} \\ \mathbb{E}_{x \sim \mathcal{D}_T}\left[\mathcal{L}\big(\phi(f(x)), \hat{f}(x)\big)\right], & \text{for hard labels.} \end{cases} \tag{2}$$

**Knowledge distillation** (KD) has been widely studied in machine learning [10, 1, 6], which transfers the knowledge from a teacher model to a student model. Model stealing attacks can be regarded as a black-box KD problem where the victim model is the *teacher* with only outputs accessible and the substitute model is the *student*. The main reason for the success of KD is the *valuable information that defines a rich similarity structure over the data* in the probability prediction [10]. However, for the hard-label setting discussed in this paper, this valuable information is lost. Inspired by KD, our method tries to dig out the hidden information in the data and models, and then transfers more knowledge to the substitute model.

**The erasing-based method**, *e.g.*, random erasing (RE) [32, 3], is currently one of the widely used data augmentation methods, which generates training images with various levels of occlusion, thereby reducing the risk of over-fitting and improving the robustness of the model. Our work is inspired by RE and designs a prior-driven erasing operation, which erases the area corresponding to the hard label to re-mine missing information.

## 3 Method

The overview of our proposed black-box dissector is shown in Fig. 2. In addition to the conventional process (*i.e.*, the transfer dataset $D_T$ constructing in step 1 and the substitute model training in the right), we introduce two key modules: a CAM-driven erasing strategy (step 2.1) and a RE-based self-KD module (step 2.2).

### 3.1 A CAM-driven erasing strategy

Since the lack of class similarity information degrades the performance of previous methods under the hard-label setting, we try to re-dig out such hidden information. Taking an example from the ILSVRC-2012 dataset for illustration as in Fig. 3. Querying the CUBS200 trained victim model with this image, we get two classes with the highest confidence score: "Anna hummingbird" (0.1364) and "Common yellowthroat" (0.1165), and show their corresponding attention map in the first column of Fig. 3. It is easy to conclude that two different attention regions response for different classes according to the attention map. When training the substitute model with the hard label "Anna hummingbird" and without the class similarity informa-

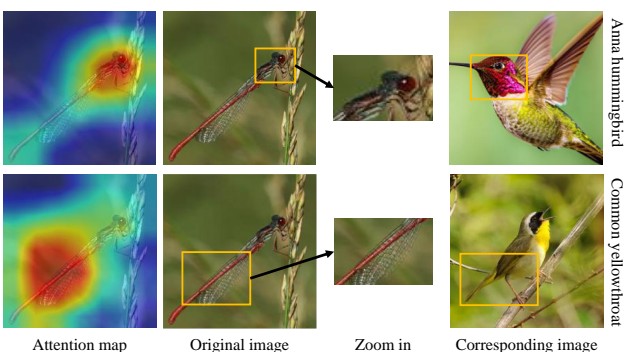

Attention map    Original image    Zoom in    Corresponding image

Figure 3: An example from the ILSVRC-2012 dataset and its attention map corresponding to two most likely class "Anna humming bird" and "Common yellow throat" on the CUBS200 trained model. The attention areas share similar visual apparent with images of "Anna humming bird" and "Common yellow throat", respectively.

tion, the model can not learn from the area related to the "Common yellowthroat" class, which means this area is wasted. To re-dig out the information about the "Common yellowthroat" class, we need to erase the impact of the "Anna hummingbird" class.

To this end, a natural idea is to erase the response area corresponding to the hard label. Since the victim model is a black-box model, we use the substitute model to approximately calculate the attention map instead. If the attention map calculated by the substitute model is inaccurate and the victim model's prediction on the erased image does not change, we can also align the attention map of two models by letting the substitute model learn the original image and the erased one simultaneously. The attention map is also a kind of supervision signal pushing two models to be similar [30]. To get the attention map, we utilize the Grad-CAM [24] in this paper. With the input image $x \in [0,1]^d$ and the trained DNN $\mathcal{F}: [0,1]^d \mapsto \mathbb{R}^N$, we let $\alpha_k^c$ denote the weight of class $c$ corresponding to the $k$-th feature map, and calculate it as $\alpha_k^c = \frac{1}{Z} \sum_i \sum_j \frac{\partial \mathcal{F}(x)^c}{\partial A_{ij}^k}$, where $Z$ is the number of pixels in the feature map, $\mathcal{F}(x)^c$ is the score of class $c$ and $A_{ij}^k$ is the value of pixel at $(i,j)$ in the $k$-th feature map. After obtaining the weights corresponding to all feature maps, the final attention map can be obtained as $S_{\text{Grad-CAM}}^c = \text{ReLU}(\sum_k \alpha_k^c A^k)$ via weighted summation.

To erase the corresponding area, inspired by [32], we define a prior-driven erasing operation as $\psi(I, P)$, shown in Alg. 1, which randomly erases a rectangle region in the image $I$ with random values while the central position of the rectangle region is randomly selected following the prior probability $P$. The prior probability $P$ is of the same size as the input image and is used to determine the probability of different pixels being erased. Here, we use the attention map from Grad-CAM as the prior. Let $x \in [0,1]^d$ denote the input image from the transfer set and $S_{\text{Grad-CAM}}^{\arg\max \hat{f}(x)}(x, \hat{f})$ denote the attention map of the substitute model $\hat{f}$. This CAM-driven erasing operation can be represented:

$$\psi\left(x, S_{\text{Grad-CAM}}^{\arg\max \hat{f}(x)}(x, \hat{f})\right). \tag{3}$$

We abbreviate it as $\psi(x, S(x, \hat{f}))$. To alleviate the impact of inaccurate CAM caused by the difference between the substitute model and the victim one, for each image, we perform this operation $N$ times ($\psi_i$ means the $i$-th erasing) and select the one with the largest difference from the original label.

**Algorithm 1:** Prior-driven Erasing Operation $\psi(I, P)$

---

**Input:** Input image $I$, prior probability $P$, image size $W$ and $H$, area of image $S$, erasing area ratio range $s_l$ and $s_h$, erasing aspect ratio range $r_1$ and $r_2$.
**Output:** Erased image $I'$.

1 $S_e \leftarrow \text{Rand}(s_l, s_h) \times S, r_e \leftarrow \text{Rand}(r_1, r_2)^1$

2 $H_e \leftarrow \sqrt{S_e \times r_e}/2, W_e \leftarrow \sqrt{\frac{S_e}{r_e}}/2$

3 $x_e, y_e$ sampled randomly according to $P$

4 $I_e \leftarrow (x_e - W_e, y_e - H_e, x_e + W_e, y_e + H_e)$

5 $I(I_e) \leftarrow \text{Rand}(0, 255)$

6 $I' \leftarrow I$

---

Such a data augment operation helps the erasing process to be more robust. We use the cross-entropy to calculate the difference between the new label and the original label, and we want to select the sample with the biggest difference. Formally, we define $\Pi(x)$ as the function to select the most different variation of image $x$:

$$
\begin{aligned}
\Pi(x) &:= \psi_k(x, S(x, \hat{f})), \\
\text{where } k &:= \underset{i \in [N]}{\arg\max} - \sum_j \phi\left(f\left(x\right)\right)_j \cdot \log\left(\hat{f}\left(\psi_i(x, S(x, \hat{f}))\right)_j\right) \\
&= \underset{i \in [N]}{\arg\max} - \log\left(\hat{f}\left(\psi_i(x, S(x, \hat{f}))\right)_{\arg\max \phi\left(f(x)\right)}\right) \\
&= \underset{i \in [N]}{\arg\min} \hat{f}\left(\psi_i(x, S(x, \hat{f}))\right)_{\arg\max \phi\left(f(x)\right)}.
\end{aligned}
\tag{4}
$$

Due to the limitation of the number of queries, we cannot query the victim model for each erased image to obtain a new label. We continuously choose the erased image with the highest substitute's confidence until reaching the budget. To measure the confidence of the model, we adopt the Maximum Softmax Probability (MSP) for its simplicity:

$$
\begin{aligned}
&\underset{x \sim \mathcal{D}_T}{\arg\max} MSP\left(\hat{f}\left(\Pi\left(x\right)\right)\right) \\
&= \underset{x \sim \mathcal{D}_T}{\arg\max} \hat{f}\left(\Pi\left(x\right)\right)_{\arg\max \hat{f}(\Pi(x))},
\end{aligned}
\tag{5}
$$

where $\mathcal{D}_T$ is the transfer set. The erased images selected in this way are most likely to change the prediction class. Then, we query the victim model to get these erased images' labels and construct an erased sample set $D_E$. Note that when the victim model's predictions on the erased images change, it means our erasing method does dig out other related class information in the sample. With the unchanged predictions, it points out the attentions of the substitute model and the victim are inconsistent. Though wrong attention areas erased, training with these samples benefits aligning the attentions of two models. As [30] stated, the attention alignment can help more powerful KD.

### 3.2 A random-erasing-based self-KD module

We also find that in training with limited hard-label OOD samples, the substitute model is likely to overfit the training set, which damages its generalization ability [15, 31]. Therefore, based on the above erasing operation, we further design a simple RE-based self-KD method to improve the generalization ability of the substitute model.

Formally, let $x \in [0, 1]^d$ denote the unlabeled input image. We perform the erasing operation with a uniform prior $U$ on it $N$ times, and then average the substitute's outputs on these erased images as the pseudo-label of the original image:

$$
y_p(x, \hat{f}) = \frac{1}{N} \sum_{i=1}^{N} \hat{f}\left(\psi_i(x, U)\right).
\tag{6}
$$

---

[1] $\text{Rand}(a, b)$ returns an evenly distributed random real number in the range of $a$ to $b$.

---

**Algorithm 2:** Black-box Dissector

**Input:** Unlabeled pool $D_U$, victim model $f$, maximum number of queries $Q$.
**Output:** Substitute model $\hat{f}$.

1   Initialize $q \leftarrow 0, D_T \leftarrow \varnothing, D_E \leftarrow \varnothing$
2   **while** $q < Q$ **do**
3     **// Step 1**
4     Select samples from $D_U$ according to budget and query $f$ to updata $D_T$
5     $q = q + budget$
6     $\mathcal{L} = \sum_{x \in D_T} \mathcal{L}'\big(\phi(f(x)), \hat{f}(x)\big)$
7     $\hat{f} \leftarrow update(\hat{f}, \mathcal{L})$
8     **// A CAM-driven erasing strategy (step 2.1)**
9     Erase samples in $D_T$ according to Eq. 4
10    Choose samples from erased samples according to Eq. 5 and budget
11    Query $f$ to get labels and updata $D_E$
12    $\mathcal{L} = \sum_{x \in D_T \cup D_E} \mathcal{L}'\big(\phi(f(x)), \hat{f}(x)\big)$
13    $\hat{f} \leftarrow update(\hat{f}, \mathcal{L})$
14    $q = q + budget$
15    **// A random-erasing-based self-KD (step 2.2)**
16    Select samples from $D_U$
17    Get pseudo-labels according to Eq. 6 and construct a pseudo-label set $D_P$
18    $\mathcal{L} = \sum_{x \in D_T \cup D_E} \mathcal{L}'\big(\phi(f(x)), \hat{f}(x)\big) + \sum_{x \in D_P} \mathcal{L}'\big(y_p(x, \hat{f}), \hat{f}(x)\big)$
19    $\hat{f} \leftarrow update(\hat{f}, \mathcal{L})$
20   **end**

---

This is a type of consistency regularization, which enforces the model to have the same predictions for the perturbed images and enhances the generalization ability. With Eq.6, we construct a new soft pseudo label set $D_P = \{(x, y_p(x, \hat{f})), \dots\}$.

With the transfer set $D_T$, the erased sample set $D_E$, and the pseudo-label set $D_P$, we train a new substitute model using the ensemble of the victim model and the previous substitute model as the teacher. Our final objective function is:

$$\min \mathcal{L} = \min \Big[ \sum_{x \in D_T \cup D_E} \mathcal{L}'\big(\phi(f(x)), \hat{f}(x)\big) + \sum_{x \in D_P} \mathcal{L}'\big(y_p(x, \hat{f}), \hat{f}(x)\big) \Big]. \tag{7}$$

where $\mathcal{L}'$ can be commonly used loss functions, *e.g.*, cross-entropy loss function.

To sum up, we built our method on the conventional process of the model stealing attack (step 1), and proposed a CAM-driven erasing strategy (step 2.1) and a RE-based self-KD module (step 2.2) unified by a novel erasing method. The former strategy digs out missing information between classes and aligns the attention while the latter module helps to mitigate overfitting and enhance the generalization. We name the whole framework as *black-box dissector* and present the algorithm detail of it in Alg. 2.

## 4 Experiment

### 4.1 Experiment settings

**Victim model.** The victim models we used (ResNet-34 [9]) are trained on four datasets, namely, CIFAR10 [16], SVHN [19], Caltech256 [7], and CUBS200 [28], and their test accuracy are $91.56\%$, $96.45\%$, $78.40\%$, and $77.10\%$, respectively. All models are trained using the SGD optimizer with momentum (of 0.5) for 200 epochs with a base learning rate of 0.1 decayed by a factor of 0.1 every 30 epochs. Following [20, 21, 34], we use the same architecture for the substitute model and will analyze the impact of different architectures in the supplementary.

**Attack dataset.** We use $1.2M$ images without labels from the ILSVRC-2012 challenge [23] as the attack dataset. In a real attack scenario, the attacker may use pictures collected from the Internet, and

Table 1: The agreement and test accuracy (in %) of each method under 30k queries. For our model, we report the average accuracy as well as the standard deviation computed over 5 runs. (**Boldface**: the best value, *italics*: the second best value.)

| Method | CIFAR10 | | SVHN | | Caltech256 | | CUBS200 | |
|---|---|---|---|---|---|---|---|---|
| | Agreement | Acc | Agreement | Acc | Agreement | Acc | Agreement | Acc |
| KnockoffNets | 75.32 | 74.44 | 85.00 | 84.50 | 57.64 | 55.28 | 30.01 | 28.03 |
| ActiveThief(Entropy) | 75.26 | 74.21 | 90.47 | 89.85 | 56.28 | 54.14 | 32.05 | 29.43 |
| ActiveThief(k-Center) | 75.71 | 74.24 | 81.45 | 80.79 | 61.19 | 58.84 | 37.68 | 34.64 |
| ActiveThief(DFAL) | 76.72 | 75.62 | 84.79 | 84.17 | 46.92 | 44.91 | 20.31 | 18.69 |
| ActiveThief(DFAL+k-Center) | 74.97 | 73.98 | 81.40 | 80.86 | 55.70 | 53.69 | 26.60 | 24.42 |
| Ours+Random | **82.14**±0.16 | **80.47**±0.02 | **92.33**±0.47 | **91.57**±0.29 | *62.15*±0.52 | *59.91*±0.58 | *38.28*±0.31 | *35.24*±0.49 |
| Ours+k-Center | *80.84*±0.21 | *79.27*±0.15 | *91.47*±0.09 | *90.68*±0.14 | **65.12**±0.56 | **62.72**±0.57 | **46.69**±0.87 | **42.91**±0.46 |

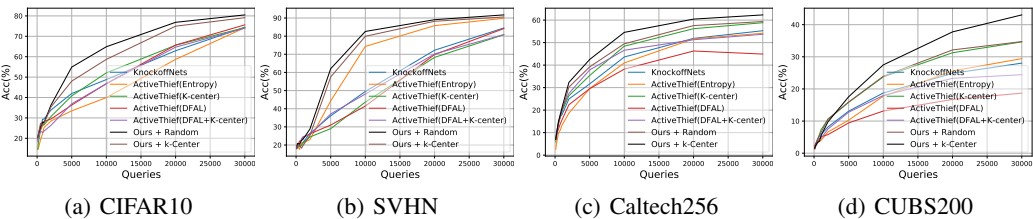

| (a) CIFAR10 | (b) SVHN | (c) Caltech256 | (d) CUBS200 |
|---|---|---|---|

Figure 4: Curves of the test accuracy versus the number of queries.

the ILSVRC-2012 dataset can simulate this scenario well. Note that we resize all images in the attack dataset to fit the size of the target datasets, which is similar to the existing setting [20, 21, 34].

**Training process.** We use the SGD optimizer with momentum (of 0.9) for 200 epochs and a base learning rate of $0.02 \times \frac{batchsize}{128}$ decayed by a factor of 0.1 every 60 epochs. The weight decay is set to $5 \times 10^{-4}$ for small datasets (CIFAR10 [16] and SVHN [19]) and 0 for others. We set up a query sequence $\{0.1K, 0.2K, 0.5K, 0.8K, 1K, 2K, 5K, 10K, 20K, 30K\}$ as the iterative maximum query budget, and stop the sampling stage whenever reaching the budget at each iteration.

**Baselines and evaluation metric.** We mainly compare our method with KnockoffNets [20] and ActiveThief [21]. Follow Jagielski et al. [12], we mainly report the test accuracy (Acc) as the evaluation metric. We also report the *Agreement* metric proposed by Pal et al. [21] which counts how often the prediction of the substitute model is the same as the victim's as a supplement.

## 4.2 Experiment results

We first report the performance of our method compared with previous methods. After that, we conduct ablation experiments to analyze the contribution of each module. Finally, we also analyze the performance of our method when encountering defense methods and real-world online APIs. More experiments (*e.g.*, adversarial attack and overfitting analysis) can be found in our supplementary.

**Effectiveness of our method.** As in Tab. 1, the test accuracy and agreement of our method are all better than the previous methods. We also plot the curves of the test accuracy versus the number of queries in Fig. 4. The performance of our method consistently outperforms other methods throughout the process. Since our method does not conflict with the previous sample selection strategy, they can be used simultaneously to further improve the performance of these attacks. Here, we take the k-Center algorithm as an example. Note that, with or without the sample selection strategy, our method beats the previous methods by a large margin. Particularly, the test accuracies of our method are 4.85%, 1.72%, 3.88%, and 8.27% higher than the previous best method, respectively. And the agreement metric shares similar results. It is also interesting that it is less necessary to use the k-Center algorithm on datasets with a small number of classes (*i.e.*, CIFAR10 and SVHN). While for the datasets with a large number of classes, the k-Center algorithm can make the selected samples better cover each class and improve the effectiveness of the method.

**Ability to evade the SOTA defense method.** The SOTA perturbation-based defense method, adaptive misinformation [13], introduces an Out-Of-Distribution (OOD) detection module based on the maximum predicted value and punishes the OOD samples with a perturbed model $f'(\cdot; \theta')$. The model $f'(\cdot; \theta')$ is trained with $\arg\min_{\theta'} \mathbb{E}_{(x,y)}[-\log(1 - f'(x; \theta')_y)]$ to minimize the probability of

Table 2: Ability to evade the state-of-the-art defense method (adaptive misinformation) on CIFAR10 dataset. The larger the threshold, the better the defence effect while the low victim model's accuracy (threshold 0 means no defence). Our method evades the defense best, and the self-KD part makes a great difference.

| Method | Threshold | | | |
|---|---|---|---|---|
| | 0 | 0.5 | 0.7 | 0.9 |
| KnockoffNets | 74.44% | 74.13% | 73.61% | 54.98% |
| ActiveThief(k-Center) | 74.24% | 69.14% | 59.78% | 50.19% |
| ActiveThief(Entropy) | 74.21% | 71.61% | 64.84% | 51.07% |
| Ours | **80.47%** | **79.95%** | **78.25%** | **74.40%** |
| Ours w/o self-KD | 79.02% | 78.66% | 73.61% | 61.81% |
| victim model | 91.56% | 91.23% | 89.10% | 85.14% |

the correct class. Finally, the output will be:

$$y' = (1 - \alpha)f(x; \theta) + (\alpha)f'(x; \theta'), \tag{8}$$

where $\alpha = 1/(1 + e^{\nu(\max f(x;\theta) - \tau)})$ with a hyper-parameter $\nu$ is the coefficient to control how much correct results will be returned, and $\tau$ is the threshold used for OOD detection. The model returns incorrect predictions for the OOD samples without having much impact on the in-distribution samples.

We choose four values of the threshold $\tau$ to compare the effects of our method with the previous methods. The threshold value of 0 means no defence. The result is shown in Tab. 2. Compared with other methods, adaptive misinformation is almost invalid to our method. Furthermore, we find that if we remove the self-KD in our method, the performance is greatly reduced. We conclude that this is because adaptive misinformation adds noise labels to the substitute model's training dataset, and self-KD can alleviate the overfitting of the substitute model to the training dataset, making this defence method not effective enough.

**Ablation study.** To evaluate the contribution of different modules in our method, we conduct the ablation study on CUBS200 dataset and plot the results in Fig. 5. If the CAM-driven erasing strategy is removed, the performance of our method will be greatly reduced, showing that it has an indispensable position in our method. We also give some visual examples in Fig. 7 to demonstrate that this strategy can help align the attention of two models. As depicted in the Fig. 7, at the beginning time, the substitute model learns the wrong attention map. Along with the iterative training stages, the attention area of the substitute model tends to fit the victim model's, which conforms to our intention. We further remove the self-KD module to evaluate its performance. It can be found from Fig. 1 and Fig. 5 that the self-KD can improve the generalization of our method and further improve the performance.

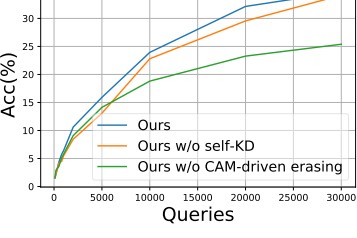

Figure 5: Ablation study on CUBS200 dataset for the contribution of the CAM-driven erasing and the self-KD in our method.

**Stealing functionality of a real-world API.** We validate our method is applicable to real-world APIs. The AWS Marketplace is an online store that provides a variety of trained ML models for users. It can only be used in the form of a black-box setting. We choose a popular model (waste classifier [2]) as the victim model. We use ILSVRC-2012 dataset as the attack dataset and choose another small public waste classifier dataset [3], containing $2,527$ images as the test dataset. As in Fig. 6, the substitute model obtained by our method achieves $12.63\%$ and $7.32\%$ improvements in test accuracy compared with two previous methods, which show our method has stronger practicality in the real world.

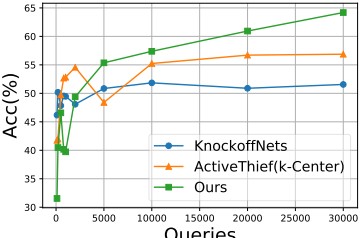

Figure 6: The experiment on AWS online API.

---

[2] https://amzn.to/3nFvA54
[3] https://github.com/garythung/trashnet

Table 3: Transferability of adversarial samples generated with PGD attack on the substitute models.

| Method | Substitute's architecture | | | | |
|---|---|---|---|---|---|
| | ResNet-34 | ResNet-18 | ResNet-50 | VGG-16 | DenseNet |
| KnockoffNets | 57.85% | 63.33% | 52.04% | 42.88% | 60.77% |
| ActiveThief(k-Center) | 57.44% | 57.90% | 57.01% | 16.49% | 60.72% |
| ActiveThief(Entropy) | 63.56% | 66.76% | 58.19% | 55.43% | 62.05% |
| Ours | **76.63%** | **74.10%** | **74.28%** | **67.03%** | **66.96%** |

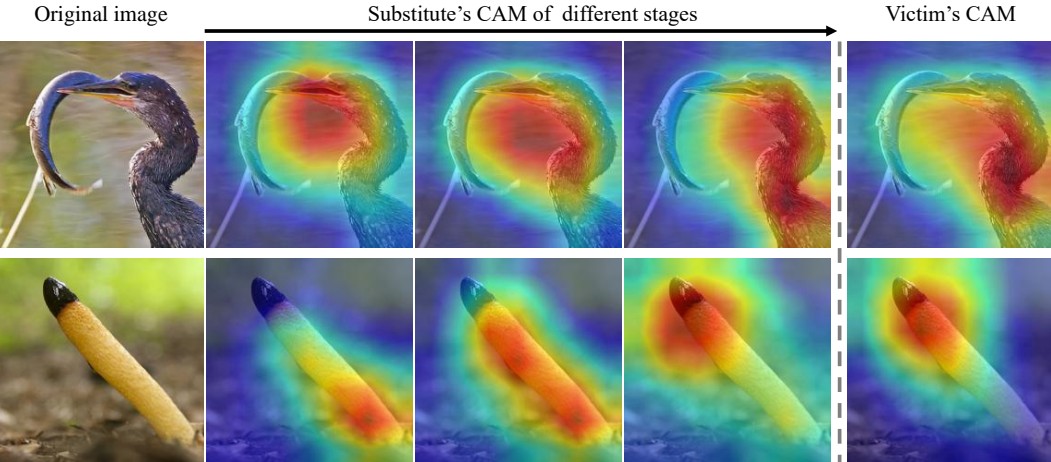

Figure 7: The visualized attention maps of the victim model and different stages substitute models using the Grad-CAM. Along with the training stages, the attention map of the substitute model tends to fit the victim model's.

**Transferability of adversarial samples.** Though with the dominant performance on a wide range of tasks, deep neural networks are shown to be vulnerable to imperceptible perturbations, *i.e.*, adversarial examples [27]. Since the model stealing attack can obtain a functionally similar substitute model, some previous works (*e.g.*, JBDA [22], DaST [34] and ActiveThief [21]) used this substitute model to generate adversarial samples and then performed the transferable adversarial attack on the victim model. We argue that a more similar substitute model leads to a more successful adversarial attacks. We test the transferability of adversarial samples on the test set of the CIFAR10 dataset. Keeping the architecture of the victim model as the ResNet-34, we evaluate the attack success rate of adversarial samples generated from different substitute models (*i.e.*, ResNet-34, ResNet-18, ResNet-50 [9], VGG-16 [26], DenseNet [11]). All adversarial samples are generated using Projected Gradient Descent (PGD) attack [18] with maximum $L_\infty$-norm of perturbations as $8/255$. As shown in Tab. 3, the adversarial samples generated by our substitute models have stronger transferability in all substitute's architectures. This again proves that our method is more practical in real-world scenarios.

## 5 Conclusion

We investigated the problem of model stealing attacks under the hard-label setting and pointed out why previous methods are not effective enough. We presented a new method, termed *black-box dissector*, which contains a CAM-driven erasing strategy and a RE-based self-KD module. We showed its superiority on four widely-used datasets and verified the effectiveness of our method with defense methods, real-world APIs, and the downstream adversarial attack. Though focusing on image data in this paper, our method is general for other tasks as long as the CAM and similar erasing method work, *e.g.*, synonym saliency words replacement for NLP tasks [4]. We believe our method can be easily extended to other fields and inspire future researchers. Model stealing attack poses a threat to the deployed machine learning models. We hope this work will draw attention to the protection of deployed models and furthermore shed more light on the attack mechanisms and prevention methods.

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
