# OpenReview forum: "Black-Box Dissector: Towards Erasing-based Hard-Label Model Stealing Attack"
_NeurIPS.cc/2021/Conference — NeurIPS 2021 Submitted_

### Official Review · Reviewer_WyuJ · 2021-06-25

**Rating:** 6
**Confidence:** 4

**Summary:**

This work presents a more effective model stealing attack using an erasing-based method which makes use of saliency maps.  The method is clever and intuitive.

**Limitations And Societal Impact:**

The authors do adequately address the limitations and impact of their work.

**Main Review:**

Thank you for the interesting work.  This paper seems to make appreciable progress on an important problem.  Below are several minor comments:

1) In general, the writing and grammar could be improved for the sake of clarity.  This is a major area for potential improvement in this paper.

2) Figure 2 is a little hard to parse.  The meaning of some of the arrows is unclear.

3)  Your victim models perform quite a bit worse than I would expect on the tasks given.  I believe hyperparameter tuning could significantly improve their performance, and maybe you could also significantly improve the performance of your attack as well with better hyperparameter tuning.

4)  It seems like the self-kd is not very important for the performance of your method.  I think you should have better justification for this component of your method.


**Time Spent Reviewing:**

3

---

> ### Author Response · Authors · 2021-08-10
> **Detailed Responses to Each Comment of Reviewer WyuJ**
>
> Thank you for the constructive comments.
>
> > In general, the writing and grammar could be improved for the sake of clarity. This is a major area for potential improvement in this paper.
>
> Thanks for the comment. We will correct grammar errors throughout the paper and ask a native English speaker to proofread our paper.
>
> > Figure 2 is a little hard to parse. The meaning of some of the arrows is unclear.
>
> We apologize for the confusion in Figure 2. Figure 2 is the overall pipeline of our algorithm, where the black arrow represents the data flow, and the red arrow represents training model with the specific dataset. In step 1, we select samples from the unlabeled pool and query the victim model to get the label. We construct a dataset $D_T$ with the samples and the labels, and then train a model with $D_T$ (the red arrow). In step 2.1, we erase each image N times according to the Grad-CAM, and select the one with the largest difference from the original image according to the substitute's output. We continuously choose the erased image with the highest confidence until reaching the budget. Then, we query the victim model to get the hard label. In step 2.2, we randomly erase each image N times, and average the substitute’s outputs to construct the soft label. After the above steps, we update the model with all the sample pairs obtained above. We will re-design the figure to make it clearer, *e.g.*, modifying the arrow for labels.
>
> > Your victim models perform quite a bit worse than I would expect on the tasks given. I believe hyperparameter tuning could significantly improve their performance, and maybe you could also significantly improve the performance of your attack as well with better hyperparameter tuning.
>
> We agree with your opinion. But we would like to clarify that for a fair comparison, we use the pre-trained victim model provided by the previous work (KnockoffNets).
> We believe that hyper-parameter optimization can improve performance both for the victim model and our method (*e.g.*, enlarging weight decay improves attack performance 1.16% as in the table below).
>
> > It seems like the self-kd is not very important for the performance of your method. I think you should have better justification for this component of your method.
>
> As per your comment, the self-KD part is not as effective as the CAM-driven erasing strategy since the latter one digs more information from the victim model. We use self-KD to improve performance mainly by preventing overfitting and miscalibration. It is particularly useful for some specific scenarios, *e.g.*, using defense methods that bring noisy labels. The results are shown in the following table, where "$2 \times$ weight decay" means enlarge of the weight decay to twice the original and the "label smoothing ($\alpha$)" means smooth the hard-label according to the hyperparameter $\alpha$. Compared to other commonly used methods including *label smoothing, data augmentation*, self-KD can bring higher performance improvement, which greatly enhances the practicability of our method.
> We will give more states in the final version.
>
> | Method | ACC |
> | :- | :-: |
> | Ours | 80.49% |
> | + $2 \times$ weight decay | 81.65% |
> | - self-KD | 79.02% |
> | - self-KD + $2 \times$ weight decay | 80.09% |
> | - self-KD + CutOut | 78.91% |
> | - self-KD + label smoothing (0.9) | 78.22% |
> | - self-KD + label smoothing (0.8) | 77.46% |

---

> > ### Comment · Reviewer_WyuJ · 2021-08-25
> > **Thanks for your response**
> >
> > Thanks for your response.  I keep my current score.

---

### Official Review · Reviewer_sWAf · 2021-07-14

**Rating:** 7
**Confidence:** 4

**Summary:**

This submission proposes a new model stealing attack method, black-box dissector, under the hard-label setting. Specifically, this paper uses CAM-driven erasing strategy and random erasing module to increase the information inquired from the victim model. Experiments are conducted on CIFAR-10, SVHN, Caltech256 and CUBS200 datasets and achieved superior performance compared to other attack methods. In addition, ablation studies are conducted to verify the effectiveness of the proposed attack.

**Limitations And Societal Impact:**

The CAM-driven erasing strategy proposed in this paper is mainly addressing the problem of information acquisition when soft labels cannot be obtained. It would be interesting to compare the attack performance between using CAM-driven erasing strategy and using soft label.

**Main Review:**

The underlying idea that gain more informative pseudo-labels by erasing response area is very insightful. Given recent upsurge in research on model and data security, the notion that model can be efficiently reproduced from APIs that output hard-labels is an emerging concern. There is also clear differentiation between black-box dissector from prior work. The experiment results also show the effectiveness of stealing attack. The reported attacks seem to significantly outperform some competitive baselines. In addition, black-box dissector can achieve high accuracy when the distributions between training dataset of victim model and attack dataset are far apart, which is of great help to other tasks such as transfer learning and domain adaptation.

Recently, there have been some new defense and detection methods against model stealing attacks [1][2]. I wonder if black-box dissector can still maintain high attack performance under these defenses.

In addition, could the authors comment on whether the label refinement [3] can improve the performance of model stealing attack if the entire attack dataset can be directly input into the victim model and get the hard-labels?

[1] P. Maini, M. Yaghini, and N. Papernot, “Dataset inference: Ownership resolution in machine learning,” in International Conference on Learning Representations, 2021.

[2] X. Wang, Y. Xiang, J. Gao, and J. Ding, “Information laundering for model privacy,” in International Conference on Learning Representations, 2021.

[3] K. Ghasedi Dizaji, A. Herandi, C. Deng, W. Cai, and H. Huang, “Deep clustering via joint convolutional autoencoder embedding and relative entropy minimization,” in Proceedings of the IEEE international conference on computer vision, 2017, pp. 5736–5745.


**Time Spent Reviewing:**

10

---

> ### Author Response · Authors · 2021-08-10
> **Detailed Responses to Each Comment of Reviewer sWAf**
>
> We thank you for your insightful comments and careful analysis of our paper.
>
> > Recently, there have been some new defense and detection methods against model stealing attacks [1][2]. I wonder if black-box dissector can still maintain high attack performance under these defenses.
>
> Following your suggestion, we evaluate the dataset inference defense [1] on the CIFAR-10 dataset. Note that this defense method is performed after training the substitute model, which will not affect the stealing process. This method protects the intellectual property and determines whether the model is stolen by measuring the amount of knowledge of the victim's dataset contained in the substitute model, which can also be regarded as a measure of the similarity between the substitute model and the victim model. Following their setting [1], we report two metrics, *i.e.*, $\Delta$ (higher is better) and p-value (lower is better). The substitute model obtained by our method has a higher $\Delta$ and a lower p-value than other methods, which also shows that our substitute model obtained is more similar to the victim model.
>
> | Method | Victim(CIFAR10 trained) | Ours | KnockoffNets | ActiveThief(k-Center) | Independent(SVHN trained) |
> | :- | :-: | :-: | :-: | :-: | :-: |
> | p-value $\downarrow$ | $7.91\times {10}^{-205}$ | $3.95 \times {10}^{-55}$ | $7\times {10}^{-47}$ | $2.36\times {10}^{-45}$ | 0.0253 |
> | $\Delta$ $\uparrow$ | 1.35 | 0.76 | 0.69 | 0.70 | 0.09 |
>
> > In addition, could the authors comment on whether the label refinement [3] can improve the performance of model stealing attack if the entire attack dataset can be directly input into the victim model and get the hard-labels?
>
> We think this is an insightful idea. With deep clustering [3], we can get the probability of the sample belonging to each cluster, and then convert it into an approximate soft label. This soft label contains richer class similarity information, which should help to improve the performance of model stealing attacks in the hard-label scenario. Furthermore, as the cost of query in model stealing is important, we believe constructing a subset of the entire dataset using active learning may help. We will cite the related works and give more discussion as the future work in the final version.
>
> > The CAM-driven erasing strategy proposed in this paper is mainly addressing the problem of information acquisition when soft labels cannot be obtained. It would be interesting to compare the attack performance between using CAM-driven erasing strategy and using soft label.
>
> Due to the page limitation, we put the attack results using the soft label in the appendix. Please refer to Table I in the appendix for more detailed results. On small datasets (CIFAR10, SVHN), the performance of our method is almost close to the one under the soft label. On large datasets (CUBS200, Caltech256), although our method has a greater improvement compared with other methods, there is still a gap compared to the soft label attack.
>
> [1] P. Maini, M. Yaghini, and N. Papernot, “Dataset inference: Ownership resolution in machine learning,” in International Conference on Learning Representations, 2021.
>
> [2] X. Wang, Y. Xiang, J. Gao, and J. Ding, “Information laundering for model privacy,” in International Conference on Learning Representations, 2021.
>
> [3] K. Ghasedi Dizaji, A. Herandi, C. Deng, W. Cai, and H. Huang, “Deep clustering via joint convolutional autoencoder embedding and relative entropy minimization,” in Proceedings of the IEEE international conference on computer vision, 2017, pp. 5736–5745.

---

### Official Review · Reviewer_ohZm · 2021-07-16

**Rating:** 6
**Confidence:** 5

**Summary:**

This paper proposes a new model stealing attack when the victim model only returns the class label to the queries instead of confidence or probability information. Prior attacks struggle in this setting to extract high-accuracy models. The proposes attack has two steps, first they query the model with a sample and probabilistically remove the input region using the GRAD-CAM of the substitute model. This aims to fit more information into a single query and simulates the prediction probabilities. Second, they use a self-knowledge distillation scheme on the substitute model to alleviate overfitting. Overall, their scheme achieves higher stolen accuracy than prior attacks, more resilient to a prior defense and allows the attacker to craft more transferable adversarial examples.

**Ethical Concerns:**

As far I see, the authors actually steal a commercially available ML model (waste classifier on AWS Marketplace). Since this is done for research, I see no harm but attacking real-world systems also have ethical and potentially legal problems. Did the authors inform the owners of this model (https://appen.com/) before conducting this attack on their system?

**Ethics Review Area:**

["Privacy and Security (e.g., consent)"]

**Limitations And Societal Impact:**

(See above)


**Main Review:**

The paper is well-written and the intuitions behind the design is pretty clear. The setting is a relevant and a realistic setting as many of the defenses rely on obfuscating/perturbing the prediction probabilities to thwart the stealing attacks. Along these lines, I would also recommend the authors to look into Prediction Poisoning defense by Orekondy et al and evaluate the efficacy against their setting where the defense does not preserve the top-1 label.

Strengths:
+ Intuitive, well-designed attack, relevant setting.
+ Circumvents the one of the prediction perturbation-based defenses (Adaptive Misinformation)

Weaknesses:
- My first main concern is the baseline attacks the authors have selected. Both ActiveThief and KnockoffNets are active learning-based and they do not modify the query samples themselves, they only select them from a larger set. The proposed attack on the other hand actively modifies the samples. I believe a more suitable baselines would be something like Prada attack that modifies the queries with perturbations to extract more information from them.

- My second main concern is the ablation study itself. The ablation in the paper completely disables one or the other step of the attack and measure the attack success. A better ablation study would replace the steps with something trivial rather than removing altogether to see whether we need a rather complicated design to begin with. For the GRAD-CAM step, experimenting with random data augmentations (for example CutOut) and for the second step using something like label smoothing to prevent overfitting to potentially noisy labels would be more informative to understand the usefulness of the design.

- I also recommend the authors to experiment with the Prediction Poisoning defense, obviously in preserving top-1 label setting, the defense would be ineffective against the proposed attack (which is a good thing), but in other settings the attack's success might be reduced.

- It is also useful to present the attack results that use the full prediction probabilities to give a sense of the upper-bound on how good a label-only attack can get.


Questions:

? Do you start from a pre-trained models before launching the model stealing attacks? This seems to be a standard practice in the prior work but I wasn't sure if it's the case here too.

? If the Adaptive Misinformation defense's OOD detector was trained on the samples generated by your attack, how would your attack fare?

==============
AFTER REBUTTAL
==============
Thanks for the extra experiments and the discussion. I'm bumping up my score, please include the detailed ablation study in the paper. I think it strengthens your story much more.

**Needs Ethics Review:**

Yes

**Time Spent Reviewing:**

3

---

> ### Author Response · Authors · 2021-08-10
> **Detailed Responses to Each Comment of Reviewer ohZm**
>
> Thank you for the constructive comments.
>
> > My first main concern is the baseline attacks the authors have selected. Both ActiveThief and KnockoffNets are active learning-based and they do not modify the query samples themselves, they only select them from a larger set. The proposed attack on the other hand actively modifies the samples. I believe a more suitable baselines would be something like Prada attack that modifies the queries with perturbations to extract more information from them.
>
> | Method | jbda | jb-self | jb-TRND | Ours |
> | :- | :-: | :-: |  :-: |  :-: |
> | ACC | 12.80% | 10.60% | 24.40% | 80.49% |
>
> Thank you for the suggestion. We argue that we choose the baselines mainly according to the setting that limits the maximum number of victim model queries. As no limitation is made in the original setting of methods modifying the queries, usually $10^{5}$ queries are needed, which is costly and unpractical. We conclude the main reason for the enormous number is that these methods modify the samples by adding adversarial noise under the black-box setting, which is difficult and challenging.
> We follow ActiveThief to set the maximum number of queries to 30K to make model stealing practical, and our method that just erases the attention map is simple enough. We evaluate two methods that modify the queries, *i.e.*, the jbda [1] and Prada (jb-self, jb-TRND) [2], on the CIFAR10 dataset, and obtained similar results as reported in [3]. The poor performance of these methods verifies the above discussion. We will give more comparisons and descriptions in the final version.
>
> [1] Nicolas Papernot, Patrick McDaniel, Ian Goodfellow, Somesh Jha, Z Berkay Celik, and Ananthram Swami. Practical black-box attacks against machine learning. In Asia CCS, 2017b.
>
> [2] Mika Juuti, Sebastian Szyller, Alexey Dmitrenko, Samuel Marchal, and N Asokan. Prada: Protecting against DNN model stealing attacks. In Euro S\&P, 2019.
>
> [3] Tribhuvanesh Orekondy, Bernt Schiele, and Mario Fritz. Prediction poisoning: Towards defenses against DNN model stealing attacks. In International Conference on Learning Representations, 2019.
>
> > My second main concern is the ablation study itself. The ablation in the paper completely disables one or the other step of the attack and measure the attack success. A better ablation study would replace the steps with something trivial rather than removing altogether to see whether we need a rather complicated design to begin with. For the GRAD-CAM step, experimenting with random data augmentations (for example CutOut) and for the second step using something like label smoothing to prevent overfitting to potentially noisy labels would be more informative to understand the usefulness of the design.
>
> Constructive suggestion. To demonstrate the usefulness of our design, we conduct a more detailed ablation study on the CIFAR10 dataset. The results are shown in the following table, where "$2 \times$ weight decay" represents the expansion of the weight decay to twice the original and the "label smoothing ($\alpha$)" means smooth the hard-label according to the hyperparameter $\alpha$. First, we replace the CAM-driven erasing with random erasing (CutOut), which brings 3.38% performance degradation. We believe that using Grad-CAM as a prior is more effective than random. Then we use data augmentation (CutOut) and label smoothing to replace the self-KD, while both show less competitive. We conclude that they destroy the information need by the CAM-driven erasing, *e.g.*, erasing the attention map or hiding information in other classes by making them equal. The result also shows the effectiveness of the self-KD module we designed.
>
> | Method | ACC |
> | :- | :-: |
> | Ours | 80.49% |
> | + $2 \times$ weight decay | 81.65% |
> | - CAM-driven erasing|76.12%|
> | replace CAM-driven erasing with random erasing (CutOut) | 77.11% |
> | - self-KD|79.02% |
> | - self-KD + $2 \times$ weight decay | 80.09% |
> | - self-KD + CutOut | 78.91% |
> | - self-KD + label smoothing (0.9) | 78.22% |
> | - self-KD + label smoothing (0.8) | 77.46% |
>
> > I also recommend the authors to experiment with the Prediction Poisoning defense, obviously in preserving top-1 label setting, the defense would be ineffective against the proposed attack (which is a good thing), but in other settings the attack's success might be reduced.
>
> Following your suggestion, we evaluate attack methods with the prediction poisoning defense on the CIFAR10 dataset. We will add these results to the subsequent version.
> Our method is less affected compared with other methods. Note that for each image, the attack baseline methods only query once, and the prediction poisoning defense perturbs the corresponding label which introduces a lot of noise. Our method erases each image for multiple labels instead. Though perturbed, our method can still dig more information. Furthermore, the self-KD module can prevent overfitting well, which also alleviates the influence of these noisy data.
>
> | Method | No defense | $\epsilon=0.5$ | $\epsilon=0.8$ |
> | :- | :-: | :-: | :-: |
> | KnockoffNets | 74.44% | 71.83% | 58.01% |
> | ActiveThief(k-Center) | 74.24% | 73.75% | 60.89% |
> | Ours | 80.47% | 80.01% | 79.23% |
> | Victim model | 91.56% | 91.56% | 89.45% |
>
> > It is also useful to present the attack results that use the full prediction probabilities to give a sense of the upper-bound on how good a label-only attack can get.
>
> Due to page limitations, we put the attack results that use the full prediction probabilities in the appendix. Please refer to Table I in the appendix for more detailed results. The performance of our method is almost close to the one under the soft label on the small datasets (CIFAR10, SVHN), while there is still a gap on large datasets (CUBS200, Caltech256).
>
> > Do you start from a pre-trained models before launching the model stealing attacks? This seems to be a standard practice in the prior work but I wasn't sure if it's the case here too.
>
> To clarify, we do not start from a pre-trained model for our method and for all baselines. We have tried with or without the pre-trained model for both soft-label and hard-label settings. The pre-trained model helps under the soft-label setting, while harms under the hard-label setting, leading to a nearly 17% performance decline for all methods compared with starting from scratch. We argue that the pre-trained model brings a strong prior with data from a different distribution, which plays the leading role under the hard-label setting since less information can be obtained from the hard labels. For example, in our method, a pre-trained model will bring a stronger attention prior, which conflicts with our goal of obtaining the victim model's attention, and then harms the performance of our method.
> We will clarify it in the final version.
>
> > If the Adaptive Misinformation defense's OOD detector was trained on the samples generated by your attack, how would your attack fare?
>
> Interesting idea. We think that our attack will fail when the OOD detector was trained on the samples generated by our method. This is because we use the ImageNet as the attack dataset, which has a very different distribution from the victim dataset, so the targeted training can make the defense method detect malicious samples easily. We believe that collecting a more similar dataset to the victim's dataset distribution helps to bypass the detector in this situation.
>
> > Ethical Concerns
>
> Thank you for pointing out the potential ethical and legal problems that we may have. We would like to clarify that it is very necessary to carry out an attack under the real online ML model, since only by doing so can we show that our attack method poses a real threat to the real world. We have contact the company with no response. If we fail to obtain permission in the future, we will hide the information about this commercial model in the final version.

---

### Official Review · Reviewer_aBhm · 2021-07-21

**Rating:** 6
**Confidence:** 3

**Summary:**

This paper proposes a new method for model stealing/knowledge distillation with hard label information only, based on a two-part modification of the standard model stealing pipeline. The standard pipeline feeds images to the target model, collects the corresponding labels, and trains a substitute model on the image-label pairs. First, the proposed method erases high-information patches from the images fed to the target model in order to collect additional (nearby) labels, providing extra information about the target model. Second, after training a first substitute model on the target images and their erased counterparts, the proposed method trains a second substitute model on both the original target data, the erased images and their labels, and "smoothed" predictions from the first substitute model in order to improve generalization.

**Main Review:**

Overall, the proposed methodology is not the simplest, but at the same time the modifications of the standard KD pipeline are creative and seem to non-negligibly improve performance on a variety of tasks. My more specific comments and questions are below:

- L149: What does "the attention map is also a kind of supervision signal pushing two models to be similar" mean? I also did not understand how this statement follows from the cited source.
- Does Section 3.1 mean that we actually train 3 substitute models throughout the process? One just on D_T to get the grad-cam, then another on D_T + D_E, then another on D_T + D_E and the smoothed pseudo labels?
- I think much of the math (e.g., the first two lines of equation 4) adds very little clarity and instead makes things more confusing by introducing additional symbols and notation. Similarly with the first line of equation 5, I think there is no reason to introduce a concept as well as an acronym (MSP) if it is only going to be used once on the immediate next line.
- The legends in Figure 4 are extremely hard to read, especially in print.
- Based on Figure 5 and the corresponding part of the results section, it seems as though the self-KD matters much less for performance than the CAM-driven erasing (which I also thought was the more well-motivated modification of the two). To what extent could the self-KD be replaced by more "conventional" ways to improve generalization and prevent overfitting like Cutout regularization, more aggressive data augmentation, higher weight decay, etc?
- The Amazon API demonstration was very interesting but lacked many experimental details---what was the setup here? What architecture was used by the proposed method vs KnockoffNets vs ActiveThief? How were parameters chosen, etc.? At face value the results are very impressive but need to be put into context with experimental details.

Overall, I think the paper showcases a creative idea that significantly improves over prior work, and presents the main ideas fairly clearly (especially in light of how complicated the method is). The clarity could be improved in places and further experimental details should be provided, and some additional results following up on the ablation study would (in my opinion) further improve the paper.

**Time Spent Reviewing:**

4

---

> ### Author Response · Authors · 2021-08-10
> **Detailed Responses to Each Comment of Reviewer aBhm**
>
> We thank the reviewer for the insightful review and valuable feedback.
>
> > L149: What does "the attention map is also a kind of supervision signal pushing two models to be similar" mean? I also did not understand how this statement follows from the cited source.
>
> Intuitively, aligning the outputs of two models to make them similar is the simplest supervision signal. The cited source uses the attention transfer to enhance the distillation effect, that is, the attention is also another kind of supervision signal. Our method performs the erasing operation based on the substitute model's attention map to mine the missing class information, which also allows the substitute model to fit the victim model's attention. This helps our method learn a more similar substitute model. For more specific discussion, please refer to L146-148 and L176-180 in the paper.
>
> > Does Section 3.1 mean that we actually train 3 substitute models throughout the process? One just on $D_T$ to get the grad-cam, then another on $D_T + D_E$, then another on $D_T + D_E$ and the smoothed pseudo labels?
>
> We agree with the reviewer that we trained three different models in three steps, and we will clarify this in subsequent versions. In terms of design, we use only one model and update it with different training datasets in different training stages. Of course, it can also be considered that different models are trained at different stages. Compared with combining the training stages and training one model directly, our method has extra training costs, while no extra space cost and victim model query costs, which are more of our concern. At the same time, our training strategy also improves the performance by 4.46% compared with one model training strategy, since the step-by-step training brings better datasets. The performance improvement deserves the extra training costs, and we will state it clearly in the final version.
>
> > I think much of the math (e.g., the first two lines of equation 4) adds very little clarity and instead makes things more confusing by introducing additional symbols and notation. Similarly with the first line of equation 5, I think there is no reason to introduce a concept as well as an acronym (MSP) if it is only going to be used once on the immediate next line.
>
> Thanks for pointing it out, and we will simplify the symbols and notations in the final version.
>
> > The legends in Figure 4 are extremely hard to read, especially in print.
>
> We apologize for the inconvenience. We will adjust the figures, *e.g.*, enlarging the size of the font, changing the style and weight of the lines, and re-organize the layout.
>
> > Based on Figure 5 and the corresponding part of the results section, it seems as though the self-KD matters much less for performance than the CAM-driven erasing (which I also thought was the more well-motivated modification of the two). To what extent could the self-KD be replaced by more "conventional" ways to improve generalization and prevent overfitting like Cutout regularization, more aggressive data augmentation, higher weight decay, etc?
>
> As per your comment, the CAM-driven erasing strategy works better since it digs more information from the victim model. We use self-KD to improve performance mainly by preventing overfitting and miscalibration, which is particularly useful for some specific scenarios like using defense methods as in Table 2. We then conduct a more detailed ablation study on the CIFAR10 dataset where the self-KD is replaced with CutOut, label smoothing, and higher weight decay. The results are shown in the following table, where "$2 \times$ weight decay" means enlarge of the weight decay to twice the original and the "label smoothing ($\alpha$)" means smooth the hard-label according to the hyperparameter $\alpha$. The CutOut and label smoothing are note suitable for model stealing attacks, and we believe that they destroy the information need by the CAM-driven erasing, *e.g.*, erasing the attention map or hiding information in other classes by making them equal. Adjusting weight decay brings performance improvement, but it is orthogonal to the self-KD. We can combine them to further improve the performance of our algorithm. We will add these in the final version.
>
> | Method | ACC |
> | :- | :-: |
> | Ours | 80.49% |
> | + $2 \times$ weight decay | 81.65% |
> | - self-KD | 79.02% |
> | - self-KD + $2 \times$ weight decay | 80.09% |
> | - self-KD + CutOut | 78.91% |
> | - self-KD + label smoothing (0.9) | 78.22% |
> | - self-KD + label smoothing (0.8) | 77.46% |
>
> > The Amazon API demonstration was very interesting but lacked many experimental details---what was the setup here? What architecture was used by the proposed method vs KnockoffNets vs ActiveThief? How were parameters chosen, etc.? At face value the results are very impressive but need to be put into context with experimental details.
>
> We apologize for not stating the details explicitly. Experiments performed on Amazon API followed the same setup as L214-218. We use the SGD optimizer with momentum (of 0.9) and weight decay (of $5 \times 10^{-4}$) for 200 epochs and a learning rate of 0.02 decayed by a factor of 0.1 every 60 epochs. The model architecture is ResNet-34. We will mention it in the final version.

---

> > ### Comment · Reviewer_aBhm · 2021-08-25
> > **Response**
> >
> > Thank you for the additional experiments and other response. I will keep my current score, though I think that the the paper still has some clarity issues that the authors should address if it gets accepted. For example, R.E.  specific points in the response:
> >
> > - L31: I think I better understand what is meant here, but the sentence in the paper is still very ambiguous. Is what you are trying to say something like "attention maps can be used as sources of additional supervision signal in distillation: encouraging a model's attention map to be similar (via a distance penalty) to that of another model also leads to the models having similar predictions."
> >
> > - I mentioned this in the first review (and I think the authors agree) but it is important to have somewhere where all of the steps are explicitly stated, e.g.
> >
> > 1. We train model A on X dataset
> > 2. We use model A to get Grad-CAM signal for each image
> > 3 ....
> >
> > This can be in the form of pseudocode, a list, table, etc. but would significantly improve (in my opinion) the presentation of the method.
> >
> > Since the authors seem willing to make such adjustments, I'll keep my score as-is.

---

> > > ### Author Response · Authors · 2021-08-25
> > > **Thanks for your response**
> > >
> > > Thanks for your further response and suggestions. We will modify the expression in the subsequent version with reference to your opinions. At the same time, we will consider showing this process in the form of detailed pseudocode to make it clearer.

---

### Review · Ethics_Reviewer_mpwL · 2021-08-09

**Recommendation:**

The concerns only lie in a small example in the paper. If the consensus is that this attack is problematic, then I think it would be sufficient to remove that example from the final paper.

**Ethical Issues:**

Yes

**Ethics Review:**

I share the authors concern regarding the attack on the AWS marketplace waste classifier since this is a commercial product. I am not an expert on the legalities around practical attacks of this form. Since the attack was done for purely academic purposes and the authors did not publish anything they learnt about the model, I think it is probably okay, although I would valuable the opinion of someone with more expertise.

---

### Decision · Program_Chairs · 2021-09-27

**Decision:**

Reject

**Comment:**

The paper studies model stealing attack in the hard-label black-box setting, where the victim model only returns the top-1 label for each query. To overcome this challenge and acquire more information, the paper designs a novel erasing-based scheme to improve the performance of model stealing attack.

This is a borderline case where the paper has pros and cons. On the positive side, the paper is proposing a novel idea that improves black-box model stealing attack by an erasing-based technique, and all the reviewers like the novelty of the paper. However, the proposed method is based on heuristics and lack fundamental theoretical justifications. In particular, it is not clear how the proposed method can overcome the challenges when Grad-CAM gives wrong or noisy input region, especially in the initial training stage. Further, the method is quite complicated (pointed out by reviewer ohZm) and there are some presentation issues (pointed out by reviewer WyuJ). After deeply looking into the paper, AC and SAC decide the reject the paper due to lack of analysis and the presentation issues. We encourage the authors to address these issues and resubmit to another top conference.